# Effects of *Piptoporus betulinus* Ethanolic Extract on the Proliferation and Viability of Melanoma Cells and Models of Their Cell Membranes

**DOI:** 10.3390/ijms232213907

**Published:** 2022-11-11

**Authors:** Justyna Bożek, Joanna Tomala, Sylwia Wójcik, Beata Kamińska, Izabella Brand, Ewa Pocheć, Elżbieta Szostak

**Affiliations:** 1Faculty of Chemistry, Jagiellonian University, 30-387 Krakow, Poland; 2Department of Chemistry, Carl von Ossietzky University of Oldenburg, 26 111 Oldenburg, Germany; 3Department of Glycoconjugate Biochemistry, Institute of Zoology and Biomedical Research, Faculty of Biology, Jagiellonian University, 30-387 Krakow, Poland

**Keywords:** *Piptoporus betulinus*, betulin, cytotoxicity, antioxidant activity, total phenolic content, Langmuir isotherm

## Abstract

*Piptoporus betulinus* is a fungus known for its medicinal properties. It possesses antimicrobial, anti-inflammatory, and anti-cancer activity. In this study, several tests were performed to evaluate the cytotoxic effect of the ethanolic extract of *Piptoporus betulinus* on two melanoma human cell lines, WM115 primary and A375 metastatic cell lines, as well as Hs27 human skin fibroblasts. The extract proved to affect cancer cells in a dose-dependent manner, and at the same time showed a low cytotoxicity towards the normal cells. The total phenolic content (TPC) was determined spectrophotometrically by the Folin-Ciocalteu method (F-C), and the potential antioxidant activity was measured by ferric-reducing antioxidant power (FRAP) assay. One of the active compounds in the extract is betulin. It was isolated and then its cytotoxic activity was compared to the results obtained from the *Piptoporus betulinus* extract. To further understand the mechanism of action of the extract’s anticancer activity, tests on model cell membranes were conducted. A model membrane of a melanoma cell was designed and consisted of 1,2-dimyristoyl-sn-glycero-3-phosphocholine, disialoganglioside-GD_1a_ and cholesterol: DMPC:GD_1a_:chol (5:2:3 mole ratio). Changes in a Langmuir monolayer were observed and described based on *Π*-*A*_mol_ isotherm and compressibility modulus changes. LB lipid bilayers were deposited on a hydrophilic gold substrate and analyzed by IR and X-ray photoelectron spectroscopy. Our study provides new data on the effect of *Piptoporus betulinus* extract on melanoma cells and its impact on the model of melanoma plasma membranes.

## 1. Introduction

As indicated by the Polish National Cancer Registry [1] and cancer incidence and mortality forecasts [2,3], in Poland the number of skin melanoma cases increases every year, and this cancer leads to the death of over half of the diagnosed patients. A similar situation can also be observed on a global scale [4], which proves the importance of the problem and the need to reach for new sources of therapeutic solutions. Patients in an advanced stage of melanoma can count on the support of treatment with methods such as immunotherapy with monoclonal antibodies that block signals from the immune checkpoints (anti-CTLA-4, PD-1 and PD-L1 inhibitors) and the use of BRAF and MEK tyrosine kinase inhibitors [5,6]. However, due to the high resistance of cells to the action of chemotherapeutic agents, alternative paths to reduce the development of melanoma are still being sought. In this context, there is a growing interest in researching natural products that can support pharmacological therapies [7,8]. A useful research object in this regard *is Piptoporus betulinus* (birch polypore; *Fomitopsis betulina*), a commonly found Poland parasitic fungus that belongs to a Basidiomycota division, which grows usually on dead birch branches and trunks [9]. Due to its pharmacological potential this fungus is so attractive to scientists that an indoor method for its cultivation was developed [10].

*Piptoporus betulinus* is known for a broad-spectrum of therapeutic activities including antiviral, antimicrobial, larvicidal, anti-inflammatory, antioxidant, immunomodulatory, and anticancer effects. Extracts of *Piptoporus betulinus* have proven cytotoxic activity against prostate cancer (DU145), [11], colon cancer (LS180) [12], thyroid carcinoma (FTC238), neuroblastoma (SK-N-AS), breast carcinoma (T47D), larynx carcinoma (Hep-2), cervix carcinoma (HeLa) [13], murine Lewis lung carcinoma (3 LL), lymphocytic leukemia (L1210) [14], and colon epithelial (HT-29) tumor cells. Preliminary studies of the cytotoxic effect of *Piptoporus betulinus* methanolic extracts on A375 melanoma cells were also conducted [11]. *Piptoporus betulinus* is a rich source of bioactive compounds [15]. It owes its antioxidant properties to the presence of β-carotene, lycopene, tocopherols, ascorbic acid, and a number of compounds from the group of flavonoids and polyphenols. In turn, the anti-inflammatory properties of *Piptoporus betulinus* can be linked to the presence of betulinic acid, betulin, polyporenic acids and lupeol [16]. These compounds have also anticancer activity similar to α-(1→3) and β-(1→6)-D-glucans, and ergosterol and indole derivatives, which were isolated from *Piptoporus betulinus* extract [17,18,19,20]. 

The main aim of the present work was to investigate the effect of ethanolic extract of *Piptoporus betulinus* on the viability of A375 metastatic melanoma cells, WM115 primary melanoma cells and Hs27 fibroblasts. Additionally, the total phenolic content and antioxidant potential of the prepared extract were determined. It was assessed whether the cytotoxic effect is a result of the activity of betulin, the compound present in the *Piptoporus betulinus* extract. It was also examined whether the components of the *Piptoporus betulinus* extract may affect the plasma cell membrane. Some tumors express high levels of tumor-associated carbohydrate antigens [21]. For example, the plasma membrane of neuroectodermal tumors including melanoma and neuroblastoma contains specific gangliosides (such as GM_2_, GM_3_, GD_2_, GD_3_, and GD_1a_) [21,22,23,24]. Their concentration is in the range of 5–20% of the total membrane lipids content. However, these gangliosides are not found in normal melanocytes [21]. Such a specific lipid composition of the cell membrane appears as an important and interesting target for the pharmacological treatment of these kinds of cancer cells. Due to the difference in response to the *Piptoporus betulinus* extract between melanoma cells and fibroblasts, we tested the changes in the properties of the plasma cell membrane models of both kinds of cells exposed to the extract.

## 2. Results and Discussion

### 2.1. Identification of Potentially Therapeutically Important Components of the Piptoporus betulinus Extract

#### 2.1.1. Determination of Total Phenolic Content (TPC) of *Piptoporus betulinus* Ethanolic Extract

Phenolic compounds are widely present in most plant and fungal tissues, and have numerous bioactive properties. They act as antioxidants and show the ability to prevent oxidative damage to organisms caused by reactive oxygen species (ROS) and/or reactive nitrogen species (RNS) [25]. In addition to antioxidant properties, flavonoids exhibit anti-inflammatory, antibacterial, antiviral, and anti-mutagenic effects [26]. Since polyphenols play such an important role in organisms, the total phenolic content (TPC) was determined in *Piptoporus betulinus* extract. For this purpose, the Folin-Ciocalteu’s assay was used [27]. TCP, expressed as milligrams of gallic acid equivalents (GAE) per 1 g of dry extract, was calculated on a base of data presented in (Appendix A) and the graph shown in (Appendix A). The concentration of the *Piptoporus betulinus* ethanolic extract used to determine TCP value was the same as the maximum extract concentration used in cell cytotoxicity, viability, and apoptosis assays. 

The regression equation of the standard curve (y = 0.9566x − 0.003, where R^2^ = 0.9984) was used to calculate total phenolic content of the *Piptoporus betulinus* ethanol extract. The determined value of TCP was 14.48 ± 0.04 mg GAE/g. Similar results were obtained for *Piptoporus* extract by Sułkowska-Ziaja et al. [28]. 

#### 2.1.2. Assessment of the Potential Antioxidant Activity of *Piptoporus betulinus* Ethanolic Extract

Antioxidants are chemical substances that can reduce oxidative stress and prevent cell damage in the body. This is because they are fantastic radical scavengers. They can act as hydrogen or electron donors and may convert hydroperoxides and hydrogen peroxide into less reactive and dangerous substances. They act as singlet oxygen quenchers, enzyme inhibitors and metal-chelating agents [29,30]. The source of antioxidants in the *Piptoporus betulinus* extract are phenolic compounds, especially flavonoids, tocopherols, ascorbic acid, β-carotene, and lycopene [28,31,32,33].

The reducing antioxidant power [34], expressed as micrograms of ascorbic acid equivalents (AEAC) per 1 g of dry extract, was determined on a base of data presented in (Appendix A) and the graph shown in (Appendix A). 

The FRAP value in AEAC was calculated; taking into account the equation of the calibration curve, the value was 18.62 ± 1.77 µg AA/g.

#### 2.1.3. Betulin Isolation

Betulin was isolated from *Piptoporus betulinus* according to the procedure described in Section 3.5. The amount of solid product obtained from 1 g of *Piptoporus betulinus* extract was 5.5 mg. The isolated white, crystalline product was examined by infrared absorption spectroscopy. The obtained experimental spectrum is presented in Figure 1 and the wavenumbers and detailed assignments of fundamental modes collected in Table 1 were compared with data presented by Falamas et al. [35]. 

The infrared spectrum of the isolated compound in the wavenumber range 3066–3772 cm^−1^ revealed the existence of a broad absorption band connected with the vibration of the OH group. At 2849, 2872, 2919, and 2957 cm^−1^, very intensive bands associated with CH_3_ and CH_2_ asymmetric and symmetric stretching vibrations also appeared. The band at 1638 cm^−1^ was observed due to a complex vibration of the CH_2_=CH-CH_3_ group connected with the R_5_ ring. Most of the bands that appeared in the fingerprint region can be assigned to bending vibrations of the OH, CH_2_, and CH_3_ groups as well as to skeletal bending modes.

### 2.2. Dose-Dependent Effect of Piptoporus betulinus and Betulin on Cell Viability, Cytotoxicity, and Apoptosis

In the next step of the study, the ethanolic extracts of *Piptoporus betulinus* were used in different cell-based assays (alamarBlue, Cell Titer, and ApoTox-Glo) performed in vitro to test their impact on cell viability (Figure 2a–c), cytotoxicity (Figure 2e), and apoptosis (Figure 2f). In addition, to compare the viability of cells treated with *Piptoporus betulinus* extract and betulin, a cell titer assay was also performed for cells cultured in the presence of the saturated ethanolic solution of betulin (Figure 2d). The saturated ethanolic solution of betulin was prepared by dissolving 2.7 mg of betulin in 0.5 mL of ethanol. 

Three different assays showed the inhibition of cell viability in the presence of *Piptoporus betulinus* extract (Figure 2a–c). A decrease in cell viability was also observed in the presence of betulin solution (Figure 2d). A375 metastatic melanoma cells were most severely affected by the *Piptoporus betulinus* extract, while the effect on Hs27 fibroblasts was significantly lower than on two melanoma cell lines. Betulin, isolated from the *Piptoporus betulinus* extract, was shown to be responsible for lowering cell viability. The dose-dependent decrease in the viability was observed for each tested cells cultured in the presence of betulin (Figure 2d).

The effect of the biologically active compounds extracted from *Piptoporus betulinus* with the use of the different solvents on cancer cell viability has been shown previously [9,11,36]. A549 human lung carcinoma, C6 rat glioma, and HT-29 human colon adenocarcinoma cell viability was significantly affected by *Piptoporus betulinus* ethyl acetate fraction in a dose–response manner, as was shown by MTT assay [36].

The results showing that the *Piptoporus betulinus* extract inhibits mainly the viability of A375 metastatic melanoma cells, but to a lesser extent WM115 primary melanoma and Hs27 fibroblasts, are promising and indicate a potentially therapeutic utility of this extract. While the reduction of the viability of all tested cell lines by the ethanolic solution of betulin is the surprising result. It can result from the higher concentration of betulin applied as a pure solution than in betulin administered in the whole extract. It seems also possible that the result observed for the *Piptoporus betulinus* extract is a cumulative effect of the compounds present in the extract with the opposite activity. Moreover, betulin applied alone is not tumor-specific, in contrast to the *Piptoporus betulinus* extract. The obtained results may seem surprising in light of the reports on the positive pharmacological effects of betulin and betulinic acid [37]. However, they confirm the conclusions drawn based on the study of betulin cytotoxicity against fish (BF-2) and mouse (NIH/3T3) fibroblasts presented in [38] by Małaczewska et al. These researchers revealed that mitochondria are very sensitive to the effect of betulin. Determined by them, betulin CC_50_ (50% cytotoxic concentration) values for tested cell lines were close to analogous IC_50_ (50% inhibitory concentration) values for cancer cells obtained by other researchers [38]. These results are interesting because, as proved Castaño and Gómez-Lechón [39], there is an almost linear correlation between IC_50_ values of fish and mammalian cells. Bearing in mind the work of Małaczewska and considering a limited number of reports on the actual effect of betulin on normal cells [40,41,42,43] we postulate that conducting further betulin safety/cytoxicity tests is necessary.

The measurement of LDH release from the cells, damaged by the active compounds present in *Piptoporus betulinus* extract, showed a statistically significant increase in LDH in the culture medium collected from above the cells treated by the highest concentration of the extract (10 μL mL^−1^). The unchanged LDH level in each cell line treated with the same ethanol volume relative to untreated cells indicates that the observed up-regulation of LDH concentrations is the result of *Piptoporus betulinus* extract activity (Figure 2e). The evaluation of caspase 3/7 activity revealed that among the dead cells, demonstrated by the LDH test (Figure 2e), are apoptotic cells (Figure 2f). WM115 primary melanoma cells are more sensitive to the cytotoxic activity of *Piptoporus betulinus* extract (Figure 2e) and undergo apoptosis (Figure 2f) more than the A375 metastatic melanoma cell line. The lowest susceptibility to *Piptoporus betulinus* extract in terms of cytotoxicity and apoptosis was shown in normal Hs27 cells (Figure 2e,f, respectively). 

A cell-specific cytotoxic effect of *Piptoporus betulinus* extract was demonstrated in the previous study by Lemieszek et al. [36]. *Piptoporus betulinus* ethyl acetate fraction, used in the highest concentration (250 μg mL^−1^), had statistically significant cytotoxic activity in an LDH assay against three rat cancer cell lines, FAO hepatoma, OLN-93 oligodendroglial, and astroglia, while two normal cell lines, HSF human skin fibroblasts and BAEC bovine aorta endothelial cells, were not sensitive to the action of this fraction [36]. Finding cytotoxic factors, including those inducing apoptosis and selectively destroying cancer cells without affecting normal cells, is the main goal of cancer research, which continues to test compounds of a natural origin. The results obtained by Lemieszek et al. [36] and our group (Figure 2) demonstrate the cancer-specific activity of *Piptoporus betulinus* extract.

Reduction or inhibition of cell viability, cytotoxic and pro-apoptotic action are not the only activities found in cell-based studies for *Piptoporus betulinus* extracts that contribute to its overall anti-cancer action. The wound healing test also showed the potential of *Piptoporus betulinus* ethyl acetate fraction to reduce rat glioma C6 cell migration [36], which is crucial to cancer cell dissemination and metastasis to distant places in the body.

### 2.3. Biomimetic Studies

#### 2.3.1. Mimicking the Cell Membrane of Melanoma

In order to test if indeed the plasma cell membrane of melanoma cells is affected by components of the *Piptoporus betulinus* extract, a model asymmetric lipid bilayer containing GD_1a_ ganglioside was fabricated (Appendix A). The inner leaflet (substrate oriented) contained DMPC and cholesterol (7:3) while the outer (interacting with the extract) contained DMPC:GD_1a_:chol (5:2:3). As a reference, a ganglioside-free, one-component DMPC bilayer, modeling the cell membrane of fibroblasts, was used. The interaction of the lipid bilayer with the *Piptoporus betulinus* extract was investigated using Langmuir isotherms, infrared spectroscopy, and x-ray photoelectron spectroscopy.

#### 2.3.2. Monolayer Studies

Figure 3 shows the Langmuir isotherm of the DMPC:GD_1a_:Chol (5:2:3) monolayer at the air|water (blue curves) and air|1% *Piptoporus betulinus* extract (red curves). 

To test the effect of 1% ethanol solution on the stability and fluidity of the lipid monolayer, the Langmuir isotherms were measured. No difference in the isotherm recorded at the air|water interface was observed. The single-component DMPC molecules in the monolayer at the air|water interface have the lift-off area (area at which the surface pressure starts to increase) of *A*_0_~1.00–1.10 nm^2^ and the limiting area (the densest packing of molecules in the monolayer) of *A*_lim_ = 0.40 nm^2^ (Appendix A) [44]. The GD_1a_ ganglioside has a large polar head group composed of six sugar residues, which determines the *A*_lim_. In the GD_1a_ monolayer the is *A*_0_ ≅ 1.5 nm^2^ and the limiting area to *A*_lim_ = 0.6 nm^2^ [45]. For the third component, cholesterol has *A*_0_ = 0.43 nm^2^ and *A*_lim_ = 0.38 nm^2^ [46]. In the DMPC:GD_1a_:chol monolayer the *A*_0_ ≅ 0.6 nm^2^ and *A*_lim_ ≅ 0.35 nm^2^ (Figure 3). Thus, in the three-component monolayer, the *A*_lim_ is smaller than for the pure cholesterol monolayer, indicating attractions between the lipids and the monolayer condensation. Interaction of the lipids at the air|liquid interface for 30 min with 1% *Piptoporus betulinus* extract affected the shape of the Langmuir isotherms (inset to Figure 3 and Appendix A). The isotherms shifted to a larger *A*_mol_ and exhibited a phase transition at 22 < *Π* < 25 mN m^−1^. The surface pressure increased slower than for the corresponding lipid monolayer at the aqueous subphase. A similar behavior was observed for interactions of the synthetic anticancer drugs doxorubicn, idarubicin and cerivestatin with phospholipid:sphyngomeylin:cholesterol model membranes [47,48].

To discuss the effect of the extract on the physical state of the lipid molecules in the monolayer film, the compressibility modulus (*K*^−1^) was calculated (Inset to Figure 3 and Appendix A) [49].
(1)$$K^{-1}={\left[-\frac{1}{A_{\mathrm{molec}}}{\left(\frac{\partial A_{\mathrm{molec}}}{\partial \Pi }\right)}_{T,P,ni}\right]}^{-1}$$

At *Π* < 40 mN m^−1^ the *K*^−1^ of DMPC was lower than 100 mN m^−1^, indicating that the lipids exist in a liquid-expanded state (inset Appendix A). At *Π* > 40 mN m^−1^ a transition to a more condensed state with *K*^−1^ ≅ 120 mN m^−1^ was observed. When the DMPC interacted with *Piptoporus betulinus* extract, the *K*^−1^ significantly dropped. Two liquid-expanded phases at 10 < *Π* < 22 and 25 < *Π* < 45 mN m^−1^ appeared in the DMPC monolayer. In the DMPC:GD_1a_:chol monolayer at the air|water interface, a transition from the liquid-expanded to a state liquid-condensed state occurred at *Π*~33–35 mN m^−1^ (Inset Figure 3). In the DMPC:GD_1a_:chol monolayer interacting with the *Piptoporus betulinus* extract, *K*^−1^ dropped to ca. 50 mN m^−1^. A phase transition at *Π* ≅ 22 mN m^−1^ was suppressed. Thus, the lipid monolayer exists in a liquid-disordered state. Concluding, certain components of the extract show surface properties, which allow them to accumulate at the interface and affect the packing of lipid molecules in the monolayer film. A shift of the area per molecule to larger values suggests that some, probably amphiphilic, components of the extract are incorporated into the monolayer. To find out the molecular aspects of this interaction, IR and XP spectroscopies were used.

#### 2.3.3. Interaction of Lipid Bilayers with *Piptoporus betulinus* Extract: A Molecular-Scale Analysis

An asymmetric model lipid membrane containing DMPC:chol in the inner and DMPC:GD_1a_:chol in the outer leaflet was transferred onto the Au surface. In the second set of experiments, the outer leaflet, prior to transfer, was left for 30 min to interact with the aqueous subphase containing 1% vol *Piptoporus betulinus* extract. IRS is an excellent analytical method for the identification of functional groups of organic molecules. The PM IRRA spectra of the lipid bilayers alone and after interaction with the *Piptoporus betulinus* extract are shown in Figure 4 and Figure 5. These spectra were compared with the ATR IR spectrum of dried on the diamond optical window *Piptoporus betulinus* extract (Figure 4a and Figure 5a).

*Piptoporus betulinus* extract in the CH stretching modes region shows IR absorption modes characteristic for aliphatic saturated and to less extent unsaturated molecules (Figure 4a). The deconvolution of this spectral region gave several overlapped IR absorption modes. A weak absorption mode at 3006 cm^−1^ was assigned to the CH stretching mode in unsaturated aliphatic molecules. The IR absorption modes below 3000 cm^−1^ originate from the methyl and methylene groups that are bound to both C and O atoms [50]. This result indicates that aliphatic hydrocarbons as well as molecules with an O atom bound to the methyl (methylene) groups, such that carbohydrates or alcohols are present in the extract.

The hydrocarbon chains in DMPC and GD_1a_ lipids in the bilayer film give well-resolved CH stretching modes of the methylene: ν_as_(CH_2_) and ν_s_(CH_2_) as well as methyl: ν_as_(CH_3_) and ν_s_(CH_3_) groups (Figure 4c). In both the DMPC and DMPC:chol—DMPC:GD_1a_:chol bilayers, the ν_as_(CH_2_) was centered at 2922 cm^−1^ while the ν_s_(CH_2_) was at 2852 cm^−1^, indicating that the hydrocarbon chains exist in a liquid phase. No change in the position of these modes was observed upon the interaction with the extract. In the spectrum of the DMPC bilayer, no IR absorption modes originating from the methyl and methylene groups attached to the O atoms were present. In the DMPC:chol—DMPC:GD_1a_:chol bilayer, these modes did not change and their positions did not overlap with the modes characteristic for the extract. It indicates that the molecules in the extract containing a significant amount of these groups (e.g., carbohydrates) do not intercalate into the bilayer.

Despite a significantly larger number of the methylene groups in the hydrocarbon chains of the lipids, the methylene-stretching modes in the lipid bilayer have comparable intensities to the methyl-stretching modes (Figure 4b,c). This is the consequence of the surface selection rule of IRRAS [51,52]. In other words, attenuation of the intensities of the methylene-stretching modes in the lipid bilayers point out that the hydrocarbon chains orient almost parallel to the membrane surface. Indeed, hydrocarbon chains in lipid bilayers containing gangliosides tend to orient normal to the bilayer plane [53]. In the bilayer interacting with the *Piptoporus betulinus* extract, an increase in the intensities of all C–CH stretching modes indicates an increase in the tilt of the chains from the surface normal. A larger inclination of the chains means a larger disruption of the packing of lipid molecules, and is in line with described above Langmuir isotherm studies.

The low-frequency spectral region provides information about the polar functional groups present both in lipids and *Piptoporus betulinus* extract (Figure 5). The dried *Piptoporus betulinus* extract shows many overlapped absorption bands in this spectral region (Figure 5a and Appendix A, and Appendix A).

The *Piptoporus betulinus* extract gives several ν(C=O) stretching modes, indicating the presence of aliphatic esters, aldehydes, ketones, and carboxylic acids, as well as conjugated and aromatic ketones and carboxylic acids. The low-frequency modes arise from molecules containing double C=C and C=N bonds and to a lesser extent aromatic molecules. The extract shows a broad, symmetric IR absorption mode centered at 3340 cm^−1^ which is assigned to the ν(OH) in the hydroxyl groups. In this spectral region, no modes arising from NH_2_-stretching modes were present. IR spectroscopy results show that the *Piptoporus betulinus* extract is composed predominantly of aliphatic molecules containing saturated bonds, which agrees with the results described in [9]. Unsaturated C=C bonds and aromatic compounds comprise a small fraction of the extract. Carboxylic acids as well as ketones, aldehydes, and alcohols belong to the major functional groups found in the *Piptoporus betulinus* extract.

The DMPC (Appendix A) and DMPC:chol—DMPC:GD_1a_:chol (Figure 5c) bilayers gave several IR absorption modes in the 1800–1400 cm^−1^ spectral region. The ester carbonyl groups in DMPC have a strong ν(C=O) mode at 1736 cm^−1^. In the three-component bilayer, this mode is overlapped with a weak ν(C=O) mode at 1714 cm^−1^, which is assigned to the carboxylic group in the sialic acid residues in GD_1a_, as well as a ν_as_(COO^−^) mode, indicating that a fraction of sialic acid residues is deprotonated [45,54]. The amide groups in GD_1a_ give the amide I mode around 1660 cm^−1^. After the interaction of the lipid bilayers with the *Piptoporus betulinus* extract, large spectral changes were observed (Figure 5b and Appendix A). Both the lipid bilayer and compounds present in the *Piptoporus betulinus* extract contribute to the measured PM IRRA spectra. The ν(C=O) mode from the ester group in DMPC is clearly seen in the spectra, however, it is overlapped with two absorption modes centered at 1714 cm^−1^ and 1704 cm^−1^. An increase in the intensity of the ν(C=O) mode in carboxylic acid residues indicates two processes in the lipid bilayer. First of all, the ester carbonyl groups in DMPC and sialic acid residues in GD_1a_ undergo some rearrangements, leading to an enhancement of this mode. Secondly, the binding of carboxylic acids residues from the extract is responsible for the appearance of the ν(C=O) mode in the 1715–1700 cm^−1^ spectral region. Two other IR absorption modes found in the extract, centered at 1670 and 1642 cm^−1^, are also present in the spectra of the lipid bilayers. The IR absorption mode at 1642 cm^−1^ is assigned to CH_2_=CH stretching mode in the R_5_ ring in betulin, one of the pharmacologically important components of the *Piptoporus betulinus* extract, as determined in 2.1. In the DMPC bilayer, they are clearly visible while in the DMPC:chol—DMPC:GD_1a_:chol bilayer they are overlapped with the IR absorption modes of the GD_1a_ molecule. Thus, betulin as well as conjugated compounds or aromatic ketones associate with the membrane.

The survey XP spectra DMPC:chol—DMPC:GD_1a_:chol bilayers revealed the presence of C, O, N and P, which belonged to the biomolecules, as well as Au from the substrate. Figure 6 shows the deconvoluted high-resolution C 1s, O 1s, N 1s, and P 2s XP spectra of the DMPC:chol—DMPC:GD_1a_:chol bilayer alone (blue) and after interaction with the *Piptoporus betulinus* extract (red curves). XPS is a surface-sensitive method, thus information about the composition and chemical environment of the bilayer surface alone and interacting with the *Piptoporus betulinus* extract is provided. In both samples, the C 1s photoelectron line was deconvoluted into four lines with full width at half maximum (fwhm) in the range of 1.05–1.45 eV. The highest content consisted of hydrocarbon chains as indicated by the strongest C 1s line at *E*_b_ = 284.9 ± 0.1 eV. The C 1s line at *E*_b_ = 286.1 ± 0.2 eV arose from the C atoms making a single bond to O and N [55,56]. Indeed, the choline and ester groups in DMPC as well as hydroxyl groups in GD_1a_ and cholesterol contribute to this mode. The C 1s line at *E*_b_ = 286.8 ± 0.2 eV originates from the C=O groups in esters and the amide of the DMPC and GD_1a_ lipids, respectively. The carboxylate and carboxylic acid groups give the C 1s line at *E*_b_ = 289.2 ± 0.2 eV. The O 1s high-resolution spectrum showed differences between the pure lipid bilayer and after its interaction with the *Piptoporus betulinus* extract. The O 1s photoelectron of the DMPC:chol – DMPC:GD_1a_:chol bilayer is composed of four overlapped lines centered at *E*_b_ = 530.8 ± 0.2 eV, *E*_b_ = 531.6 ± 0.2 eV, *E*_b_ = 532.6 ± 0.2 eV and *E*_b_ = 533.5 ± 0.2 eV. The assignment of these lines is shown in Figure 6b.

Upon interaction of the lipids with the *Piptoporus betulinus* extract, the low binding energy O 1s line could not be deconvoluted in the XP spectrum (Figure 6f). This photoelectron line is assigned to P=O^−^ and COO^−^ groups in DMPC and GD_1a_, respectively [55,56]. The other three O 1s photoelectron lines are present in both bilayers. The position of these lines does not change, but their relative intensities change (Table 2).

An increase in the relative intensity of the O 1s line at *E*_b_ = 533.5 eV indicates an increased content of protonated carboxylic acid residues and/or water bound to the bilayer. In contrast, the content of surface phosphate groups in phospholipids decreased. Thus, the composition and chemical environment of the bilayer after interaction with the extract changed.

The high resolution XP N 1s spectrum of the DMPC:chol—DMPC:GD_1a_:chol bilayer contains two lines at *E*_b_ = 399.6 ± 0.2 eV, representing the amide groups in GD_1a,_ and *E*_b_ = 402.4 ± 0.2 eV of the choline group in DMPC (Figure 6c) [55,56]. After interaction with the *Piptoporus betulinus* extract, in 70% of cases, a third new N 1s line at a low *E*_b_ = 396.2 ± 0.2 eV appeared in the spectra (Figure 6g). It is assigned to N=C (N≡N, N≡C) species. The P 2s line at *E*_b_ = 191.5 ± 0.2 eV arises from the phosphate group in DMPC. The atomic ratio of the [N 1s]_402_:[P 2s] is 1:1.32 and deviates from the expected 1:1 ratio. The expected thickness of the lipid bilayer is close to 6 nm; thus, the N 1s signal from the choline group [(CH_3_)-N^+^] in the inner, Au-facing, leaflet may be attenuated compared to the P 2s signal of the phosphate group.

The total elemental composition of the DMPC:chol—DMPC:GD_1a_:chol bilayer alone and after interaction with the *Piptoporus betulinus* extract is summarized in Table 2.

Even though after the interaction of the membrane with the extract a new species containing N was detected, an overall decrease in the content of N and P atoms was observed. This result suggests a decrease in the content of DMPC molecules in the bilayer. An increase in the content of C atoms, in particular of the C 1s line at *E*_b_ = 284.9 eV of the saturated hydrocarbons, was noted. No C 1s line *E*_b_ < 284.5 eV could be deconvoluted in the XP spectra, indicating that unsaturated, aromatic compounds are absent, according to the resolution of XPS. The overall content of O changed a little after interaction with the extract, despite large changes in the chemical environment of oxygen present in the sample.

Concluding, spectroscopic studies allowed for the determination of different functional groups from the extract interacting with the lipid bilayers. The IR spectroscopy gave clear evidence that the carboxylic acid groups in molecules containing predominantly saturated hydrocarbons have an affinity to the lipid bilayers. An increase in the relative intensities of the C 1s line at *E*_b_ = 284.9 eV and the O 1s line at *E*_b_ = 533.5 eV, observed in XPS experiments, indicates that carboxylic acids, such as those identified in the extract, multicyclic acids (e.g., betulinic acid or poliporenic acid) as well as fatty acids (e.g., palmitoyl acid, stearic acid, oleic acid, or linoleic acid) [57], interact with the DMPC and DMP:chol-DMPC:GD_1a_:chol model membranes. The isotherm studies indicated the increased fluidity of the lipid bilayers interacting with the extract. In addition, the XPS results showed that only in the three-component bilayer the content of N and P in the bilayer upon interaction with the extract decreased. This may have occurred for two reasons: (i) removal of DMPC molecules from the bilayer and/or (ii) change in the bilayer composition and thickness caused by an accumulation of the molecules from the extract, so that the N and P signals from the inner, Au-facing leaflet are attenuated. Fatty acids are surface-active, amphiphilic molecules that, in small quantities, are found in cell membranes [58]. Their accumulation in the lipid bilayer affects the membrane fluidity, phase behavior, permeability, membrane fusion, lateral pressure, and flip-flop dynamics [59]. Summarizing, the surface active components of the extract function as detergents, removing the liquid phospholipids and incorporating the amphiphilic acids into the bilayer.

However, the lipids present in the liquid-ordered (rafts) phase undergo reorganizations which include size and composition changes. Observed in this study were changes in the membrane fluidity, as well as an increased disorder of chain packing, ratio of DMPC to GD_1a_ and cholesterol content, which indicate that amphiphilic molecules such as fatty acids present in the extract were incorporated into the lipid bilayers, affecting the packing of the lipid molecules.

It is well-documented in the literature that cholesterol stabilizes the liquid-ordered phase of lipids, participating in the formation of lipid rafts [60]. Plant or fungal sterols affect the phase separation, packing, size and stability of liquid-ordered domains rich in gangliosides [61,62]. Betulinic and poliporenic acids sterols are present in the *Piptoporus betulinus* extract [57]. The incorporation of these compounds into the bilayer is in line with the appearance of the IR absorption modes in 1700–1600 cm^−1^. Since the content of carboxylic acids and hydrocarbons in the bilayer interacting with the DMPC:chol—DMPC:GD_1a_:chol bilayer increases, these molecules may also associate with the lipid membrane, affecting its fluidity and the packing of lipid molecules. Indeed, polycyclic anticancer drugs have confirmed affinity to the lipid membranes, being in line with the above-described results. 

## 3. Materials and Methods

### 3.1. Chemicals

1,2-dimyristoyl-sn-glycero-3-phosphocholine (DMPC), cholesterol and disialoganglioside-GD_1a_ (porcine brain, diammonium salt) (GD_1a_) were purchased from Avanti Polar Lipids (Acton, MA, USA). Lipids were used as received. Sodium hydroxide, potassium bromide, Folin-Ciocalteu reagent, gallic acid, potassium hexacyanoferrate(III), iron(III) chloride hexahydrate, sodium dodecyl sulfate (SDS), hydrochloric acid, methanol, ethanol, 2-propanol, toluene, acetone, and chloroform were purchased from Sigma Aldrich (Steinheim, Germany). Microscope glass slides (VWR International BVBA, Leuven, Belgium) were cut into pieces of 1.0 × 2.5 cm^2^ and rinsed with water and 2-propanol. Afterward, the glass slides were dried in a stream of argon. On the cleaned glass surface, 0.7 nm of adhesive Cr and 150 nm Au layers were evaporated using a Tectra MinoCoater instrument (Tectra GmbH, Frankfurt/Main, Germany). Before each LB transfer, the slides were rinsed with water and ethanol, dried with Ar, and placed in a UV/ozone cleaner (Bioforce Nanoscience Inc., Ames, IA, USA) for 10 min.

### 3.2. Ethanol Extraction of Piptoporus betulinus

*Piptoporus betulinus* was purchased from Aromatika (Hajnówka, Poland). Raw material (Birch hub) was dried and diced. The place of the origin of the raw material was the Polish Białowieza Forest. A total of 14.3 g of *Piptoporus betulinus* was extracted for 4 h with ethanol (v = 300 mL) in a Soxhlet apparatus. The resulting yellow–brown extract was filtered through filter paper. Then, 50 mL of such obtained solution was evaporated to dryness under vacuum and then solid residue was weighed. The rest of the extract was stored at 4 °C and protected from light.

### 3.3. Spectrophotometric Determination of the Total Phenolic Content (TPC)

The total content of phenolic compounds was determined by the Folin-Ciocalteu method (F-C), and the used reference substance was gallic acid. The tests were carried out on the extract obtained directly from the extraction process without its further dilution. F-C is a colorimetric method based on the reversible reduction of molybdenum (VI) to molybdenum(V) by phenols. The reaction produces a blue compound that exhibits absorption at 745–750 nm. The intensity of the absorption at this wavelength is proportional to the phenol concentration. The measurements were performed using a SHIMADZU UV-1280 spectrometer (Kyoto, Japan). Briefly, a calibration curve was prepared by using an appropriately water-diluted solution of gallic acid in an ethanol/water mixture (1:9 *v*/*v*) with a starting concentration of 5 mg mL^−1^. A standard solution with a given concentration was mixed with (F-C) reagent and deionized water in a volume ratio (1:5:79). The resulting mixture was vortexed and then left to stand for 4 min at room temperature. After this time, 7.5% solution of sodium carbonate was added. The volume of this solution was 15 times greater than the gallic acid standard. The obtained mixture was vortexed again and placed in a thermostat set to 40 °C for 30 min. Finally, the absorbance of such prepared sample was measured at 765 nm against a reagent blank. An analogous determination procedure was carried out by replacing the standard gallic acid solution with the tested *Piptoporus betulinus* ethanolic extract or 96% ethanol, respectively.

### 3.4. Ferric Reducing Antioxidant Power (FRAP) Assay

The reduction of ferric ions by *Piptoporus betulinus* ethanolic extract was evaluated according to the procedure described in [28]. The reference substance was ascorbic acid (AA). The tests, similar to the F-C method, were carried out on the *Piptoporus betulinus* extract obtained directly from the extraction process without its further dilution. Briefly, a calibration curve was prepared by using an appropriately ethanol-diluted solution of ascorbic acid with a starting concentration of 1 mg mL^−1^. Prepared standard solutions of ascorbic acid were mixed with deionized water, 1 M HCl, 1% K_3_[Fe(CN)_6_], 1% SDS, and 0.2% FeCl_3_, respectively. The volume ratio of mixed solutions was 10:63:2:15:5:5. This mixture was vortexed and then left to stand for 30 min in the dark. After this time, the absorbance was measured by using a SHIMADZU UV-1280 spectrometer at 750 nm. An analogous determination procedure was carried out by replacing the standard ascorbic acid solution with the tested *Piptoporus betulinus* ethanolic extract of a given concentration or 96% ethanol, respectively.

### 3.5. Isolation and Purification of Betulin

A total of 50 g of *Piptoporus betulinus* was extracted for 4 h with 200 mL of ethanol in a Soxhlet apparatus. After cooling, the resulting extract was filtered and again placed in a round-bottomed flask. The solvent was evaporated until the final volume of the solution was about 20 mL, and then 50 mL of a 5% methanolic NaOH solution was added. The resulting mixture was refluxed for 10 h and then placed in the refrigerator for 12 h. The precipitated betulin was filtered off, washed 5 times with distilled water, and then crystallized, first from propan-2-ol and then from a mixture of toluene–methanol (1:4) and acetone.

### 3.6. Infrared Spectroscopy Measurement

The IR spectrum of betulin was recorded in the 4000–400 cm^−1^ spectral region using a VERTEX 70 v apparatus (Bruker, Ettlingen, Germany). The spectra were collected using the KBr pellet method. A total of 64 scans were accumulated with a resolution of 2 cm^−1^, first from propan-2-ol and then from a mixture of toluene–methanol (1:4) and acetone.

### 3.7. Cell Lines and Culture Conditions

The cytotoxic assays were performed on three human cell lines: Hs27 derived from normal cells and two cell lines with different stages of melanoma progression. The human foreskin fibroblast Hs27 line was kindly provided by the Jagiellonian University Medical College in Krakow. WM115 cell line was obtained from the ESTDAB Melanoma Cell Bank (Tübingen; Germany). Human malignant melanoma A375 cells were purchased from the American Type Culture Collection (ATCC) (Rockville, MD, USA).

Hs27 cells were grown in DMEM high glucose medium (Gibco, 41965–039, Paisley, UK) and the melanoma cells were cultured in RPMI 1640 medium (Gibco, 72400–054) supplemented with 10% heat-inactivated fetal bovine serum (FBS; Gibco, 10270–106), and 100 units mL^−1^ penicillin and 100 μg mL^−1^ streptomycin (Gibco, 151140–122). All cultures were maintained in a 37 °C humidified incubator with 5% CO_2_ (Forma Steri-Cycle i160, Thermo Scientific, Rockford, IL, USA).

When confluency was around 80–90%, cells were trypsinized using trypsin–EDTA solution (Sigma-Aldrich, T4049, St. Louis, MO, USA) and transferred into new cell culture dishes (TPP, Trasadingen, Switzerland) with a fresh media. All cell lines used in experiments were *Mycoplasma*-negative, which was determined by a MycoAlertTM Mycoplasma Detection Kit (Lonza, LT07–318, Walkersville, MD, USA).

### 3.8. Cell Cytotoxicity, Viability, and Apoptosis Assays

The effect of *Piptoporus betulinus* extract and the saturated ethanolic solution of betulin on Hs27, WM115, and A375 cells was tested by cell viability assays: colorimetric CellTiter (G4000, Promega, Madison, WI, USA) and fluorometric alamarBlue (DAL1025, Thermo Fisher, Waltham, MA, USA), cytotoxic lactate dehydrogenase (LDH) activity test (MAK066, Sigma-Aldrich, St. Louis, MO, USA), as well as ApoTox-Glo assay (G6320, Promega, Madison, WI, USA) were used to analyze both cell viability and apoptosis.

Cells, detached by trypsinization and counted after trypan blue staining (Bio-Rad, TC-10, Hercules, CA, USA), were seeded in 96-well flat-bottomed plates (1–2 × 10^4^ cells per well), either transparent (TPP) or black (Invitrogen, Paisley, UK), and incubated overnight. Cells in each plate were treated for 24 h with different concentrations of *Piptoporus betulinus* ethanolic extract (1, 2.5, 5, and 10 μL mL^−1^), the saturated ethanolic betulin solution (0.01, 0.05, 0.25, 1, 5, 7.5, and 10 μL mL^−1^), or the highest concentration (10 μL mL^−1^) of ethanol (EtOH) (Figure 7).

#### 3.8.1. AlamarBlue Assay

The measurement of cell viability by alamarBlue reagent was based on the reduction of non-fluorescent resazurin to highly fluorescent resorufin, whose spectrum was then detected (Figure 7). The culture medium was aspirated from above the cells, and the wells were washed twice in phosphate-buffered saline (PBS) (Gibco, 14190-136). The alamarBlue reagent was diluted tenfold in PBS and added to the wells (100 μL/well). Cells were incubated in a CO_2_ incubator for 3–4 h, and then fluorescence was measured at 535 nm (Ex)/595 nm (Em) using Infinite Pro 200 microplate reader (Tecan, Männedorf, Switzerland) against alamarBlue reagent diluted in PBS.

#### 3.8.2. CellTiter Assay

The CellTiter assay determines the number of viable cells based on the quantitative measurement of formazan, which is produced by metabolically active cells (Figure 7). After careful removal of the culture medium, the cells were washed twice in PBS. The CellTiter Dye reagent, diluted 1:9 in PBS, was added to each well (100 μL per well) and the plates were incubated in a CO_2_ incubator for 3–4 h. The cells were permeabilized by a stop mix reagent (100 μL per well) in the dark overnight at room temperature (RT). The absorbance of each well was measured at a wavelength of 570 nm (Infinite Pro 200, Tecan) against the CellTiter Dye reagent diluted in PBS.

#### 3.8.3. ApoTox-Glo Assay

Cell viability and apoptosis were determined using the ApoTox-Glo Assay within a single test. Cleaving of a peptide, GF–AFC (glycyl–phenylalanyl–aminofluorocoumarin), by cellular proteases resulted in a fluorescent signal proportional to the number of viable cells, while a chemiluminescence signal from a luminogenic substrate, cleaved by caspase-3/7, was measured to detect apoptosis (Figure 7). GF–AFC substrate (20 µL) was added to each well containing 100 µL of culture medium, the plate was incubated in CO_2_ incubator for 30 min, and fluorescence was measured at 400 nm (Ex)/505 nm (Em) (Infinite Pro 200, Tecan). Then, Caspase-Glo^®^ 3/7 Reagent (100 µL) was added to the wells, the plate was incubated for 30 min at RT, and chemiluminescence was measured (Tecan).

#### 3.8.4. LDH Activity

The activity of LDH, released from the damaged cells to culture medium (conditioned medium, CM), was determined to assess the cytotoxic effect of *Piptoporus betulinus* and betulin on fibroblasts and melanoma cells. CM from the above cells (50 µL) was transferred to a new plate and a reaction mix (50 µL) consisting of a substrate mix diluted in the assay buffer was added to the medium. The LDH assay was run also for the positive control and NADH standard according to the manufacturer protocol. The absorbance was measured at 450 nm on Expert Plus microplate reader (ASYS/Hitech, Eugendorf, Austria). After the initial read, the plate was incubated at 37 °C, and read continuously every five minutes until the absorbance value of the most active sample was greater than the value of the highest NADH standard (12.5 nmole/well). In the linear range of the standard curve, these absorption values were obtained after 5 and 15 min of incubation with *Piptoporus betulinus* extract and betulin solution, respectively.

#### 3.8.5. Statistical Analysis

The fluorescence intensity, absorbance or chemiluminescence values for the treated cells were referred to the untreated cells or the cells cultured in the presence of the highest EtOH concentration (10 µL mL^−1^). The results on the graphs are represented as percentage of control ± SD. The statistical analysis was performed in the GraphPad Prism 9 (GraphPad Software, Inc., San Diego, CA, USA). The Shapiro–Wilk test was used to verify a normal distribution and then the statistical significance in comparison to untreated cells (*) was determined by a one-way ANOVA followed by Dunnett’s multiple comparisons test and marked on the graphs as follows: * *p* ≤ 0.05; 2* *p* ≤ 0.01; 3* *p* ≤ 0.001; 4* *p* ≤ 0.0001.

### 3.9. Langmuir–Blodgett Transfer

Before each experiment, fresh lipid solutions were prepared. DMPC and cholesterol were dissolved in CHCl_3_. GD_1a_ was dissolved in CHCl_3_: CH_3_OH in a 9:1 volume ratio. In one set of experiments, pure DMPC lipid monolayers and bilayers were prepared at the air|water and Au substrate, respectively. The concentration of DMPC was 1 mg mL^−1^. In the second set of experiments, transversely asymmetric three-component lipid bilayers were prepared at the Au surface. The inner, Au-facing leaflet contained DMPC:chol (7:3 mole ratio) while the outer leaflet contained DMPC:GD_1a_:chol (5:2:3 mole ratio) lipids. In each lipid mixture, the total lipid concentration was 1 μmol mL^−1^. Using a microsyringe (Hamilton, Reno, NV, USA), several μLs of the lipid solution were placed at the liquid|air interface of the Langmuir trough (KSV, Helsinki, Finland). The subphase contained either water (resistivity 18.2 MΩ cm, PureLab Classic, Elga LabWater, Celle, Germany) or 1% vol ethanolic *Piptoporus betulinus* extract in water. Surface pressure (*Π*) vs. area per molecule (*A*_mol_) isotherms were recorded in the KSV LB mini trough (KSV Ltd., Helsinki, Finland) equipped with two hydrophilic barriers. Surface pressure was recorded as a function of the area per molecule. The accuracy of measurements was ±0.02 nm^2^ for *A*_mol_ and ±0.1 mN m^−1^ for *Π*. LB and LS transfers were used to prepare symmetric DMPC and asymmetric lipid bilayers containing DMPC:chol in the inner (substrate-oriented) and DMPC:GD_1a_:Chol in the outer (solution-oriented) leaflets. The lipid monolayers were transferred onto a gold surface from the air|liquid interface at *Π* = 30 mN m^−1^. First, the inner leaflet was transferred from the aqueous subphase by a vertical LB withdrawing at a speed of 35 mm min^−1^. The transfer ratio was 1.10 ± 0.15. After the transfer, the PE monolayer was left for 1 h for drying. Next, the lipids of the second leaflet were compressed to *Π* = 30 mN m^−1^ and transferred via the Langmuir–Schaefer method onto the Au slide covered by the inner leaflet monolayer. The gold substrate was covered by Y-type lipid bilayers. Planar lipid bilayers were dried for at least 4 h before use in further experiments.

### 3.10. Polarization Modulation Infrared Reflection Absorption Spectroscopy

PM IRRA spectra were measured with a Vertex 70 spectrometer and an external reflection setup (Bruker, Ettlingen, Germany) containing a photoelastic modulator with the frequency of 50 kHz and a demodulator PMA 50 (Hinds Instruments, Hillsboro, OR, USA). The PM IRRA spectra were recorded at the air|Au interface. The half-wave retardation of the photoelastic modulator was set to maximum efficiency at 2900 cm^−1^ for the investigation of the CH-stretching modes and at 1600 cm^−1^ for the studies of the IR absorption modes in the polar head group region of the lipids. A total of 400 scans with a resolution of 4 cm^−1^ were recorded. The angle of incidence of the IR beam was set to 80°. PM IRRA spectra were processed using OPUS v5.5 software (Bruker, Ettlingen, Germany). The spectra depicted in this manuscript show the absorbance of the organic molecules present on the metal surface as a function of wavenumbers, as calculated after background subtraction by spline-interpolation and normalization of the raw PM IRRAS signal.

### 3.11. Attenuated Total Reflection Infrared Spectroscopy

Attenuated total reflection infrared spectroscopy (ATR IRS) spectra were recorded by co-adding 64 scans with a nominal resolution of 4 cm^−1^ of the dried *Piptoporus betulinus* extract on a single reflection silicon prism using an MVP-Pro ATR unit (Harrick Scientific Products, Inc., Pleasantville, NY, USA) and the Bruker Vertex 70 spectrometer.

### 3.12. X-ray Photoelectron Spectroscopy

X-ray photoelectron spectra were measured from the LB lipid bilayers deposited on a hydrophilic Au|Cr|glass substrate, using an ESCALAB 250 Xi spectrometer (Thermo Fisher Scientific, East Grinstead, UK). The XPS instrument was operated using a monochromatic Al K_α_ X-ray radiation source (*h*ν = 1486.6 eV, 288 W). LB monolayers on an Au slide were grounded by connecting carbon tape to the surface and the sample holder. The acquisition time was kept rather low to avoid any radiation damage to the sample. The survey spectra were collected at pass energy *E*_pass_ = 100 eV, step size Δ*E* 0 1 eV, and dwell time *τ* = 50 ms. The high-resolution C 1s, N 1s, O 1s, P2p, P2s and Au 4f were collected at *E*_pass_ = 20 eV, Δ*E* = 0.1 eV, and *τ* = 50 ms. The binding energy scale was referenced to the Au 4f_7/2_ line at *E*_B_ = 84.0 eV.

## 4. Conclusions

Nature is probably going to be a source of inspiration for those looking for new drugs for a long time to come. This is evidenced by the fact that of all the anti-cancer drugs approved by the FDA for use between 1981 and 2014, about 46% originated directly from or were inspired by nature [63]. The common feature of the compounds that were extracted from plants and exhibit cytotoxic activity toward cancer cells is their antioxidant potential. Among the compounds with proven anti-cancer activity, we can often find polyphenols and triterpenes, for example betulin. In the present study the cytotoxic potential of *Piptoporus betulinus* ethanoic extract was assessed. For this purpose, the antioxidant activity and total phenolic content of the extract were examined. Our research revealed relatively high levels of polyphenols (*Grifola frondosa* 6.36, *Auricularia auricula-judae* 7.93, *Agrocybe aegerita* 11.8 mg, and *Piptoporus betulinus* 14.48 GAE/g sample extract), compared to other fungi with medical applications.

Results of the FRAP assay proved that antioxidant bioactive compounds can be found in the *Piptoporus betulinus* extract. These biomolecules play an important role in scavenging free radicals in the body thus protecting cells from damage. Although it is believed that the activity of betulin is similar to betulinic acid, as shown by Dubinin et al., this substance can increase the generation of ROS in mitochondria and contribute to the development of oxidative stress in cells [64].

To sum up the results of biological research, our study confirmed the anti-cancer activity of *Piptoporus betulinus* ethanolic extract in relation to melanoma cells with the deepest cytotoxic effect against metastatic cells and the negligible impact on fibroblasts viability. The use of the purified betulin showed a dose–response cytotoxic effect, but it was not cancer-specific. It needs further study to assess if the observed effect resulted from the higher betulin concentration or/and the cumulative activity of the different compounds richly presented in the extract, e.g., phenols and amphiphilic acids detected in our study. The results of our research constitute a premise for the assessment of *Piptoporus betulinus* anti-cancer activity in other models, including animal models.

We showed also that the cell-specific response for *Piptoporus betulinus* ethanolic extract may result from the specific composition of plasma membrane lipids. Conducted research revealed that fatty acids and sterols found in the extract interact with the model membranes. This is not specific to the composition of the membrane, indicating that both control DMPC and melanomas mimicking DMPC:chol—DMPC:GD_1a_:chol bilayers are destabilized by the extract, which acts as a detergent. However, in the three-component DMPC:chol—DMPC:GD_1a_:chol bilayer, modeling melanoma plasma membranes, structural reorganizations in the membrane seem to destabilize the phospholipid-rich bilayer fragment and lead to a removal of the DMPC molecules from the membrane. XPS analysis indicates the removal of DMPC molecules from the bilayer or change in the bilayer composition and thickness caused by interaction with the compounds present in the extract. Thus, intercalation of carboxylic acids and fatty acids from the extract into the membrane and disruption of its composition and structure may be responsible for the cytotoxic behavior of the extract towards cancer cells. Further studies on model membranes mimicking, more exactly, fibroblast and different melanoma cells are still required.

## Figures and Tables

**Figure 1 ijms-23-13907-f001:**
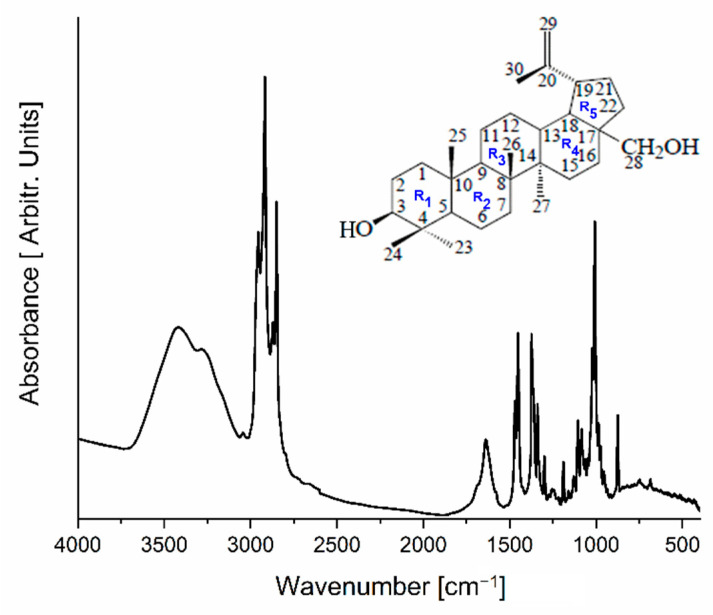
The FT-IR spectrum of a solid product isolated from the *Piptoporus betulinus* extract.

**Figure 2 ijms-23-13907-f002:**
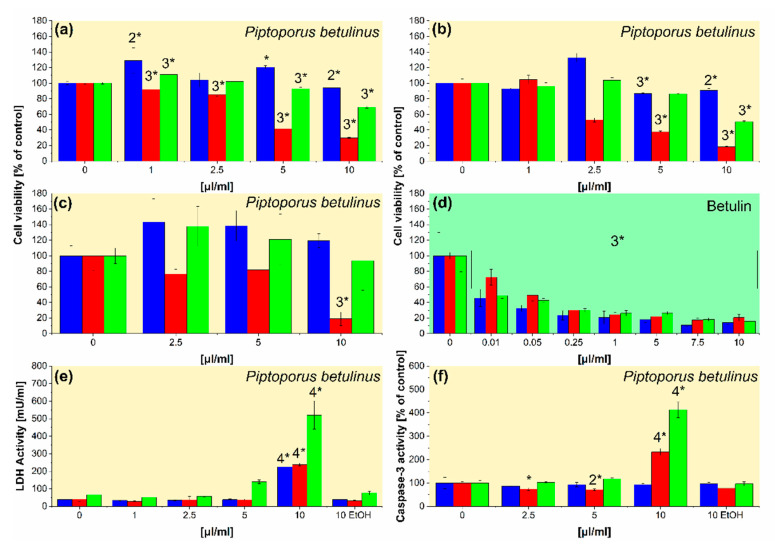
The effect of *Piptoporus betulinus* and betulin on Hs27 (blue columns), A375 (red columns), and WM115 (green columns) in terms of (**a**–**d**) cell viability, (**e**) cytotoxicity, and (**f**) apoptosis. The cells, cultured in 96-well plates, were treated with the ethanolic extracts of *Piptoporus betulinus* (1, 2.5, 5, and 10 μL mL^−1^), betulin solution (0.01, 0.05, 0.25, 1, 5, 7.5, and 10 μL mL^−1^), or the highest concentration (10 μL mL^−1^) of ethanol (EtOH) for 24 h. Cell viability was evaluated by (**a**) alamarBlue, (**b**,**d**) CellTiter, and (**c**) ApoTox-Glo assays; cell cytotoxicity was analyzed by (**e**) LDH test; apoptosis was determined by (**f**) ApoTox-Glo assay. The results are presented as means ± SD of three repetitions, and asterisked values differ significantly between treated and untreated cells at * *p* < 0.05, 2* *p* < 0.01, 3* *p* < 0.001, and 4* *p* < 0.0001 as determined by one-way ANOVA followed by Dunnett’s multiple comparisons test.

**Figure 3 ijms-23-13907-f003:**
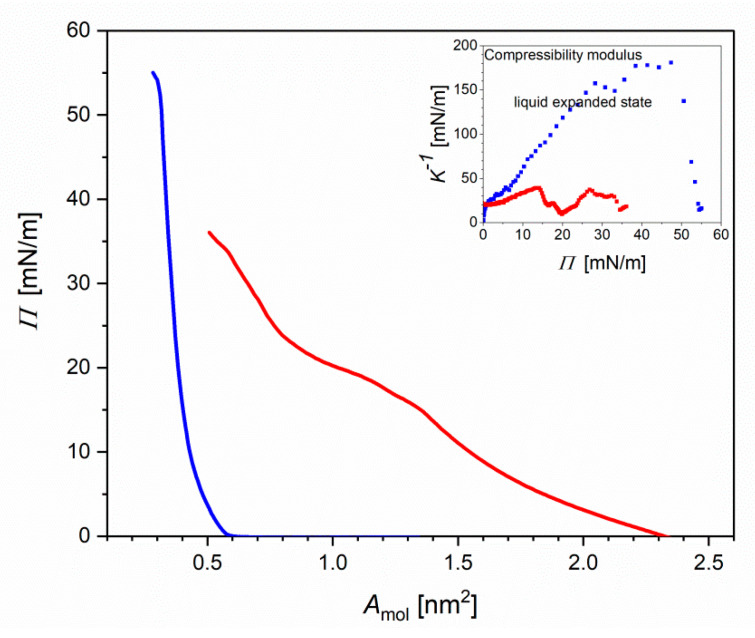
Surface pressure *Π* versus area per molecule (*A*_mol_) isotherms of the DMPC:GD_1a_:chol (5:2:3) lipid monolayer on the aqueous (blue) and 1% aq. *Piptoporus betulinus* extract after 30 min of incubation (red). Inset: Compressibility modulus versus *Π* of the lipid monolayer on the aqueous (blue) and 1% vol *Piptoporus betulinus* ethanolic extract in water after 30 min of incubation (red).

**Figure 4 ijms-23-13907-f004:**
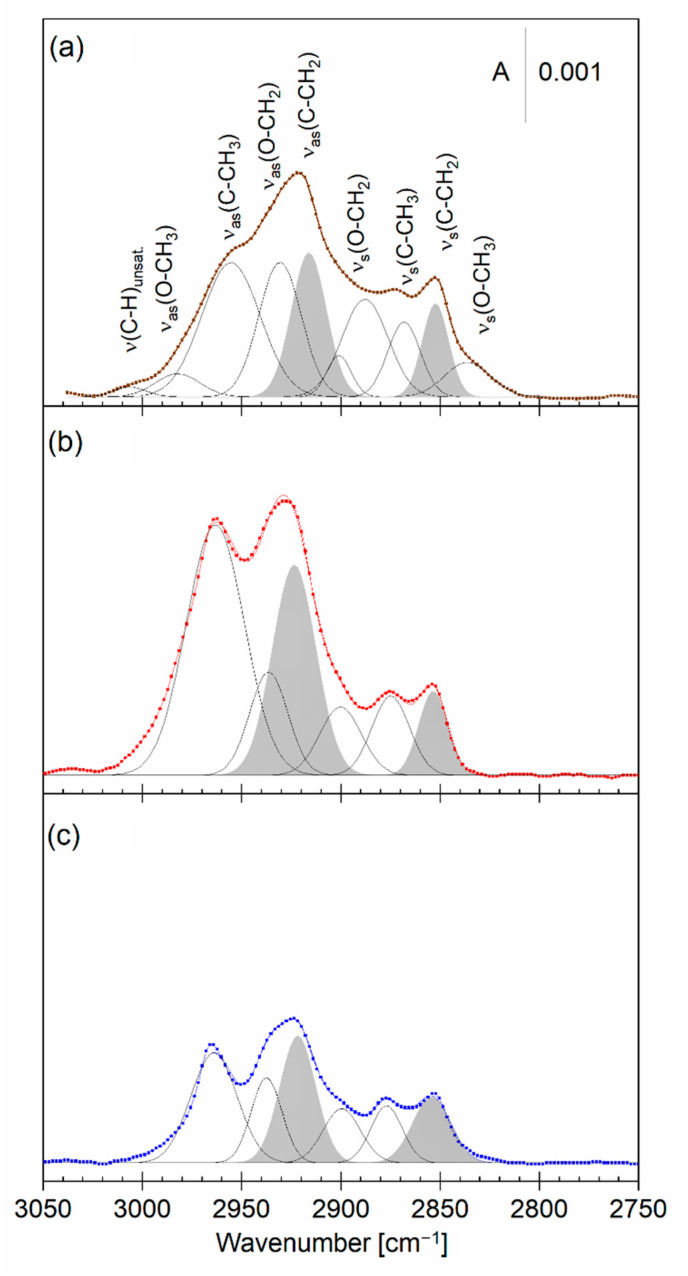
(**a**) Deconvoluted ATR IR spectrum (brown line) of a dried *Piptoporus betulinus* extract with assignment of the deconvoluted bands and (**b**,**c**) deconvoluted PM IRRA spectra (red and blue lines) of the asymmetric DMPC:chol—DMPC:GD_1a_:chol bilayer on the Au surface transferred from (**b**) 1% vol *Piptoporus betulinus* extract in the aqueous subphase (red) and (**c**) aqueous subphase (blue) and in the CH stretching modes region. A—absorbance.

**Figure 5 ijms-23-13907-f005:**
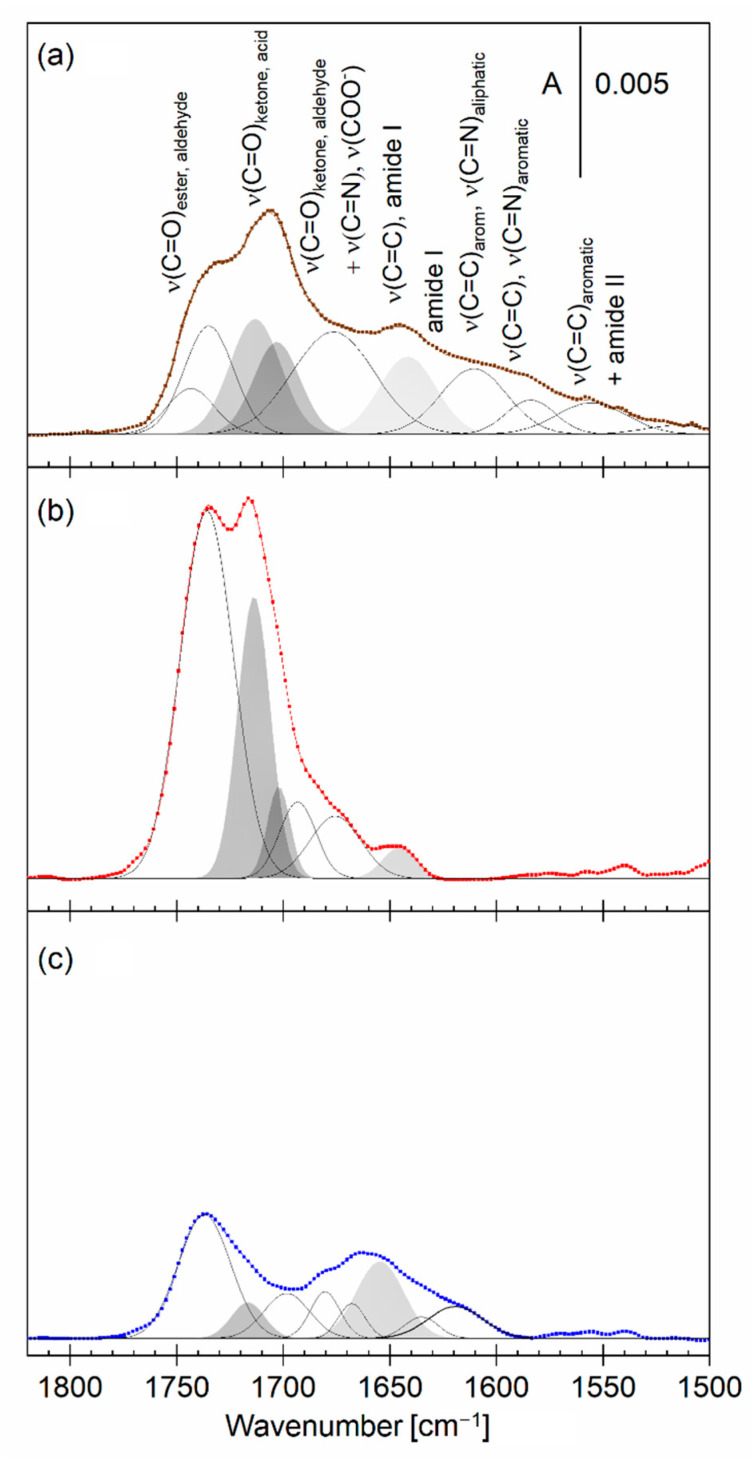
(**a**) Deconvoluted ATR IR spectrum (brown line) of a dried extract from *Piptoporus betulinus* with assignment of the deconvoluted bands and (**b**,**c**) deconvoluted PM IRRA spectra (red and blue lines) of the asymmetric DMPC:chol—DMPC:GD_1a_:chol bilayer on the Au surface transferred from (**b**) 1% vol *Piptoporus betulinus* extract in the aqueous subphase (red) and (**c**) aqueous (blue) subphase in the 1820–1500 cm^−1^ spectral region. A — absorbance.

**Figure 6 ijms-23-13907-f006:**
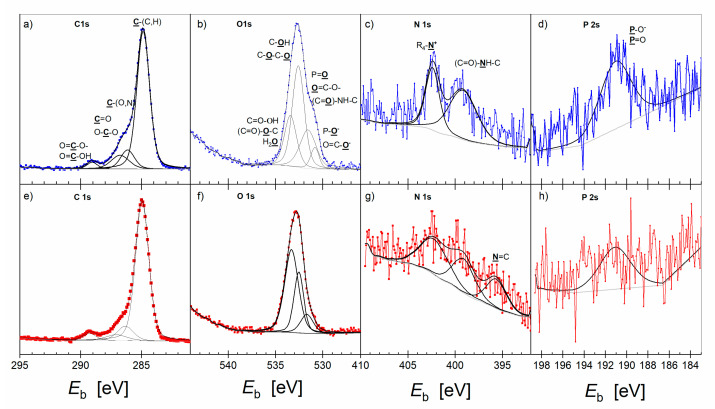
High-resolution C 1s (**a**,**e**), O 1s (**b**,**f**), N 1s (**c**,**g**) and P 2s (**d**,**h**) XP spectra of the asymmetric DMPC:chol—DMPC:GD_1a_:chol bilayer on the Au surface transferred from aqueous (**a**–**d**; blue) and 1% vol *Piptoporus betulinus* extract aqueous subphase (**e**–**h**; red). Assignment to each element to different chemical environments is given in each figure.

**Figure 7 ijms-23-13907-f007:**
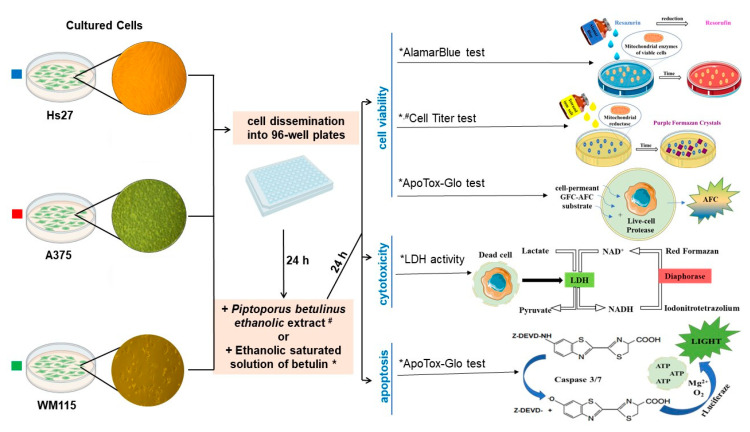
Scheme of in vitro cell-based assays used to determine Hs27, A375, and WM115 cell viability, cytotoxicity, and apoptosis in the presence of *Piptoporus betulinus* extact (tests marked in the figure with a hash mark (#)), and cell viability in the presence of betulin solution (tests marked in the figure with an asterisk (*)), and their principles.

**Table 1 ijms-23-13907-t001:** Comparison of band positions of the solid product obtained during isolation of betulin from *Piptoporus betulinus* with an experimental IR spectrum of betulin (vw—very weak, w-weak, sh—shoulder, m—medium, s—strong, vs—very strong, br—broad, ν—stretching, ρ—rocking, δ—bending, τ—twisting; ω—wagging, as—asymmetric, s—symmetric).

IR_exp_ in: KBr	Betulin Literature IR Data [35]	Vibrational Assignment [35]
685 w	685 w	ρ(C_30_H_2_) + τ(C_29_H_2_) + δ (C_18_C_19_C_21_) + δ(C_18_H) + δ (C_4_C_3O_)
748 br, w	713 vw	ρ (CH_2_)R_5_ + ρ(OC_28_H) + ν(C_17_C_28_) + R_4_ breathing + ρ(C_15_H_2_)
873 m	875 s	ω(C_29_H_2_)
917 w	913 w	ρ(CH_3_)R_1_ + ν_as_(C_23_C_4_C_24_) + ρ(C_26_H_2_)
934 w	931 w	ρ(C_23_H_2_) + δ(OH)R_1_ + δ(CH) + ρ(C_26_H_3_) + ρ(C_29_H_2_) + ν(C_20_C_30_)
946 sh 955 sh		δ(OH)R_1_ + ρ(C_24_H_2_) + δ(CH)R_2_,R_3_,R_5_ + ν_s_(C_5_C_6_C_7_) + δ(CH)R_1_,R_2_,R_3_
973 sh	972 m	ν(C_15_C_16_) + ν_as_(C_19_C_21_C_22_) + ρ(CH_3_) + δ(OH)R_1_ + ρ(C_28_H_2_)
986 sh	984 w	δ(OH) + ν(OC_3_) + ρ(CH_3_)R_1_ + δ(CH) + ρ(C_28_H_2_) + ρ(C_29_H_2_)
1007 s	1006 vs	ν(C_28_O) + δ(C_19_H) + ρ(CH_3_CH_2_)
1024 s 1035 sh	1035 s	ν(C_3_C_2_) + δ(OH) + δ(CH)R_1_,R_2_,R_3_ + ρ(CH_3_)
1063 sh	1068 sh	ν(C_28_O) + δ(C_19_H) + ρ(CH_3_CH_2_)
1082 m	1083 m	τ(C_22_H_2_) + δ(C_21_H) + δ(C_18_H) + δ(C_13_H) + ρ(C_16_H_2_) + ρ(CH_3_)R_3_,R_2_
1105 m	1104 m	δ(OH) + ω(C_2_H_2_) + δ(CH) + ω(C_6_H_2_) + ρ(C_25_H_2_) + ν(C_10_C_5_) + ρ(C_28_H_2_) + ν(C_18_C_17_)
1130 w	1136 vw	δ(OH)R_1_ + δ(CH)R_1_ + ρ(CH_3_)R_1_ + ν_as_(C_24_C_4_C_5_)
1159 w	1159 vw	δ(OH)R_5_ + ω(C_28_H_2_) + δ(C_19_H) + τ(C_16_H_2_) + ρ(C_29_H_2_)+ δ(C_29_H) + ν(C_20_C_30_)
1189 m	1188 m	δ(OH)R_5_ + δ(C_28_H) + δ(C_19_H) + ρ(C_29_H_2_) + ν(C_20_C_30_) + δ(C_30_H) + δ(C_3_H) + δ(OH)R_1_
1217 vw	1211 w	δ(OH)R_1_ + δ(C_2_H) + τ (C_28_H_2_) + ω(C_22_H_2_)
1241 vw	1243 w	δ(OH) + τ(C_28_H_2_) + δ(CH)R_2_,R_3_,R_4_,R_5_
1250 vw	1260 vw	ω(CH_2_)R_3_,R_4_,R_5_ + τ(C_22_H_2_) + δ(C_18_H) + δ(C_13_H) + δ(C_9_H) + δ(C_3_H)
1276 vw	1281 vw	δ(CH)
1299 m	1298 m	δ(CH) + ν(C_19_C_20_) + ν(C_18_C_19_) + δ(C_29_H) + ν(C_14_C_13_)
1339 m	1338 m	δ(CH_2_CH_3_) + δ(C_18_H) + δ(C_13_H) + δ(CH_3_)R_3_
1363 sh	1360 sh	δ(CH_3_)R_2_,R_3_ + δ(C_13_H) + δ(C_18_H)
1374 s	1373 s	δ(CH_3_)R_2_,R_3_ + δ(C_13_H) + δ(C_18_H)
1451 s	1450 s	δ(CH_3_) + δ(CH_2_)
1470 m	1463 sh 1485 sh	δ(CH_3_)R_1_,R_2_ + δ(C_2_H_2_) + δ(C_6_H_2_) δ(CH_3_)R_1_,R_2_ + δ(C_6_H_2_)
1638 m	1639 m	δ(C_29_H_2_) + ν(C_20_C_29_) + δ(C_19_C_20_H)
2849 s	2849 sh	ν(C_5_H) + ν(C_3_H)
2872 vs	2866 vs	ν(C_2_H_2_) + ν(C_1_H_2_) + ν(C_9_H)
2919 vs	2909 sh 2929 vs	ν_as_(C_28_H_2_) + ν(C_21_H_2_) + ν(C_15_H) + ν(C_13_H) + ν(C_12_H)
2957 vs	2947 sh	ν_as_(C_23_H_2_) + ν_as_(C_24_H_2_) + ν_as_(C_27_H_2_)
3045 w 3276 sh	3073 w 3212 m 3362 s	ν_as_(C_29_H_2_) + ν(OH)
3422 br	3419 s 3474 s	ν(OH)R_5_ + ν(OH) of adsorbed water

**Table 2 ijms-23-13907-t002:** Element composition of the pure DMPC:chol—DMPC:GD_1a_:chol bilayer and after its interaction with the *Piptoporus betulinus* extract.

Bilayer/Element	C 1s		O 1s		N 1s		P 2s	
	*E*_b/_eV	% Atom.	*E*_b/_eV	% Atom.	*E*_b/_eV	% Atom.	*E* _b/_ ^eV^	% Atom.
DMPC:chol DMPC:GD_1a_:chol	284.9 286.1 286.8 289.2	57.0 8.5 8.0 3.0	530.8 531.6 532.6 533.5	3.0 4.0 6.0 4.0	396.0 399.6 402.4	0.00 1.75 2.25	191.5	2.5
TOTAL		76.5		17.0		4.00		2.5
DMPC:chol DMPC:GD_1a_:chol + *Piptoporus betulinus* extract	284.9 286.1 286.8 289.2	65.0 8.0 4.0 3.0	530.8 531.6 532.6 533.5	0.0 2.0 4.5 9.0	396.0 399.6 402.4	0.75 1.00 1.25	191.5	1.5
TOTAL		80.0		15.5		3.00		1.5

## Data Availability

Data is contained within this article.

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
