# Peer review of "Effects of Piptoporus betulinus Ethanolic Extract on the Proliferation and Viability of Melanoma Cells and Models of Their Cell Membranes"

_ijms, 2022, doi:10.3390/ijms232213907_

Round 1

Reviewer 1 Report

The work submitted by Justyna Bożek and coworkers addresses the potentiality of  Piptoporus betulinus extracts for the therapy of skin melanoma. The results may be of interest to the audience. Some suggestions are addressed below, which may improve the reading of the article and highlight the significance of the work.

Figures 1 and 2 show calibration curves. I suggest moving it to the supplementary data file.

The results exposed in figure 4 are confusing. Three different assays were performed to study the cytotoxicity of Piptoporus betulinus extracts in addition to LDL and caspase 3 activity. In contrast, only one cytotoxic assay was performed for student Betulin. I cannot follow the rationale for those studies. Please justify.

Figures 5 and S2. The authors show compression isotherms of the ternary mixture DMPC:GD1a:Chol or DMPC with or without the presence of the Piptoporus betulinus extracts. While the surface adsorption of amphiphilic components of the extracts is obvious in both systems, it is not so if they affect more the "melanoma mimicking membrane" than the control PC membrane. Can the authors comment on this point?

From the results shown in Figure 8 and Table 2, the authors state that "an overall decrease in the content of N and P atoms was observed". The authors comment on the possible selective removal of DMPC, which does not justify the decrease in the N signal. It would be interesting if the authors directly address the possibility that the extract was acting as a detergent and removing the lipids from the gold surface. 

Reviewer 2 Report

The manuscript “Effects of Piptoporus betulinus ethanolic extract on the proliferation and viability of melanoma cells and models of their cell membranes” describes interesting study that tested effect of Piptoporus betulinus on melanoma cells. In the study active component, betulin is extracted and also tested for cytotoxic effect on melanoma cells. Additionally, effects of Piptoporus betulinus was tested on melanoma cell membrane.

The study is interesting however, some corrections need to be done.

 Comments

Page 2, lines 64-71. The aim of the study should be in the separated paragraph and merged with the next paragraph (lines 72-80).

Page 2, lines 83-94. Please omit this. This paragraph is just repetition of the aim of the study and is not necessary.

Page 3, Figure 1. Figure 1 should be presented as supplementary material (together with Table S1). The calibration curve is not significant result of this study, and do not need to be presented in the main text. Here should be only stated the level of TCP in the prepared extract. Just for comparison to the results of cytotoxicity, what was the concentration of the extract of which TCP was assessed?

Page 3, lines 126-130. It is not necessary to explain the procedure in so many details. This need to be stated in Materials and Method section. TCP and FRAP assay are well known methods and such detailed explanations are not needed. Why betulin is not tested for FRAP activity?

Page 4. Figure 2. As Figure 1, Figure 2 should be transferred to supplementary material. The authors should focus on the main results.  

Page 6, line 162, “its component” should be omitted.

Page 6, lines 166-174. There is a problem with Figure 4. It is stated that same assays (three cell viability assays, cytotoxicity and apoptosis assays) were performed for Piptoporus betulinus extract solutions and also for betulin solutions. But, on Figure 4 is only one (4d) result for betulin present. If other tests are not performed for betulin, please state clearly that only one test is performed.

Page 6, lines 180-181, here should be stated that only one test, presented on Figure 4d, is performed for betulin solutions.

Page 15, lines 512-513. Here should be stated that for testing original extract (not diluted) is used.

Page 16, lines 525-526. Same comment.

Page 16, lines 534. Some text is missing.

Page 16, line 572. It is stated “saturated ethanolic betulin solution”. Will be nice to know how saturated ethanolic solutions of betulin is prepared, what was the mass of betulin and the volume of ethanol added.

Page 17, Figure 9. Figure 9 is not correct if Piptoporus betulinus extract and betulin solution are not tested using all assays.

Round 2

Reviewer 2 Report

The authors answered to all raised questions and improved the manuscript.